# Vitamin D and Hospital Admission in Older Adults: A Prospective Association

**DOI:** 10.3390/nu13020616

**Published:** 2021-02-14

**Authors:** Avril Beirne, Kevin McCarroll, James Bernard Walsh, Miriam Casey, Eamon Laird, Helene McNulty, Mary Ward, Leane Hoey, Anne M. Molloy, Martin Healy, Catherine Hughes, Sean Strain, Conal Cunningham

**Affiliations:** 1Mercers Institute for Research on Ageing, St James’s Hospital, D08 NYH1 Dublin, Ireland; 2Department of Gerontology, St James’s Hospital, D08 NYH1 Dublin, Ireland; kmccarroll@stjames.ie (K.M.); jbwash@stjames.ie (J.B.W.); mcasey@stjames.ie (M.C.); ccunningham@stjames.ie (C.C.); 3School of Medicine, Trinity College, D02PN40 Dublin, Ireland; lairdea@tcd.ie (E.L.); AMOLLOY@tcd.ie (A.M.M.); 4Nutrition Innovation Centre for Food and Health, School of Biomedical Sciences, University of Ulster, Coleraine, Northern Ireland BT52 1SA, UK; h.mcnulty@ulster.ac.uk (H.M.); mw.ward@ulster.ac.uk (M.W.); l.hoey@ulster.ac.uk (L.H.); c.hughes@ulster.ac.uk (C.H.); jj.strain@ulster.ac.uk (S.S.); 5Department of Biochemistry, St James’s Hospital, D08 NYH1 Dublin, Ireland; mhealy@stjames.ie

**Keywords:** vitamin D, vitamin D deficiency, hospitalisation, hospital admission, emergency department attendance, resource utilisation

## Abstract

The health effects of vitamin D are well documented, with increasing evidence of its roles beyond bone. There is, however, little evidence of the effects of vitamin D on hospitalisation among older adults. This study aimed to prospectively determine the relationship of vitamin D status in older adults with hospital admission and emergency department (ED) attendance. Trinity University of Ulster Department of Agriculture (TUDA) is a large cross-sectional study of older adults with a community population from three disease-defined cohorts (cognitive dysfunction, hypertension, and osteoporosis). Participants included in this analysis were recruited between 2008 and 2012. ED and hospital admission data were gathered from the date of TUDA participation until June 2013, with a mean follow up of 3.6 years. Of the 3093 participants, 1577 (50.9%) attended the ED during the period of follow-up. Attendees had lower mean serum 25(OH)D concentrations than non-attendees (59.1 vs. 70.6 nmol/L). Fully adjusted models showed an inverse association between vitamin D and ED attendance (Hazard Ratio (HR) 0.996; 95% Confidence Interval (CI) 0.995–0.998; *p* < 0.001). A total of 1269 participants (41%) were admitted to hospital during the follow-up. Those admitted had lower mean vitamin D concentrations (58.4 vs. 69.3 nmol/L, *p* < 0.001). In fully adjusted models, higher vitamin D was inversely associated with hospital admission (HR 0.996; 95% CI 0.994–0.998; *p* < 0.001) and length of stay (LOS) (β = −0.95, *p* = 0.006). This study showed independent prospective associations between vitamin D deficiency and increased hospitalisation by older adults. The need for further evaluation of current recommendations in relation to vitamin D supplementation, with consideration beyond bone health, is warranted and should focus on randomised controlled trials.

## 1. Introduction

Increased use of healthcare resources by older populations is a complex issue in modern healthcare, associated with negative outcomes in relation to function, independent living, and quality of life. Presentation to the emergency department (ED) for older adults can be associated with poorer outcomes of functional decline, institutionalisation, and mortality. Hospital admission rates and length of stay (LOS) are other measures used to quantify resource use. Reduction in LOS is considered a potential strategy to optimise resource consumption and reduce health care costs.

Many studies have tried to identify those at increased risk of ED attendance and hospitalisation. The effects of vitamin D deficiency on bone are well-established, causing rickets and osteomalacia, osteoporosis, falls [1], and fractures [2]. Given postulated multisystem effects of vitamin D beyond bone, studies have analysed the association between vitamin D and specific conditions such as infections, sepsis [3,4,5,6], exacerbations of chronic illnesses, and functional outcomes following falls and fractures [7,8,9,10]. Vitamin D also appears to be associated with cardiovascular risk through promoting atherosclerosis [11] and has been linked to inflammation, increased platelet volume, and vascular stiffness [12,13]. Vitamin D deficiency stimulates systemic and vascular inflammation, enabling atherogenesis [14]. Hypertension is also associated with lack of vitamin D, due to activation of the Renin Angiotensin Aldosterone (RAA) system, causing endothelial dysfunction. Evidence also supports a role of 25-hydroxyvitamin D (25(OH)D) in cognitive function, with a number of studies assessing the association between cognition and vitamin D [15,16,17]. The effects of vitamin D on the brain and on cognition are potentially mediated through a number of mechanisms, including effects on neurotransmitters, neuro-inflammation, the vitamin D receptor, and genetic effects [18,19,20,21,22].

Associations between vitamin D deficiency and adverse outcomes have been conflicting, depending on study populations.

Previous studies demonstrated that frailty is associated with increased ED attendance and increased rates of hospitalisation and with higher healthcare costs [10,11]. A number of studies have evaluated the relationship between vitamin D and frailty and have found an association between deficient levels and frailty measures. [12,13,14].

Given the paucity of evidence for an association between serum 25(OH)D and hospitalisation, the aim of this study was to evaluate the prospective relationship between serum vitamin D and resource use including ED attendances, hospital admission, and LOS in older Irish community-dwelling adults.

## 2. Materials and Methods

This analysis was conducted in 3093 participants recruited to the Trinity-Ulster, Department of Agriculture (TUDA) study through St James’s Hospital (SJH), Dublin. The TUDA study comprised a community-dwelling population of Irish adults aged 60 years and older. It was a cross-sectional study designed to create a genotype/phenotype database for three population cohorts based on three disease states: cognition, bone health, and hypertension. This current study examined the cognition and bone health cohorts as they were originally recruited from SJH. The full methodology is published elsewhere [23].

### 2.1. Vitamin D Measurements

Samples for 25(OH)D analysis (the recognised marker for the nutritional status of vitamin D) included total serum 25(OH)D (D2 and D3) concentrations, which were quantified by a fully validated method (Chromsystems Instruments and Chemicals GmbH, Grafelfing, Germany; MassCrom25-OH-VitaminD3/D2) using liquid chromatography–tandem mass spectrometry (API 4000; AB SCIEX, Chesire, United Kingdom) and batch-analysed in the Biochemistry Department of St. James’s Hospital (accredited to International Organisation for Standardisation (ISO) 15,189 standard). The quality and accuracy of the method were continuously monitored by the use of internal quality controls, participation in the vitamin D external quality assessment scheme (DEQAS), and the use of the National Institute of Standards and Technology (NIST)972 D standard reference material.

The recognised cut-offs for vitamin D, as defined by the Institute of Medicine (IOM) of deficiency: <30 nmol/L, insufficiency: 30–50 nmol/L, and sufficiency: >50 nmol/L were adopted in this analysis.

### 2.2. ED Attendance

Details relating to ED attendance were accessed through the SJH Therefore system, an electronic record of all ED attendance. Information was gathered from the date of TUDA participation until June 2013, including the number of and reason for ED attendance. The reason for ED attendance was established from individual records, extracting diagnosis through predefined keywords: falls, fracture, cardiopulmonary, stroke/transient ischaemic attack (TIA), gastrointestinal bleed, other/unclear, planned.

### 2.3. Hospital Admissions

Details of hospitalisation were accessed through the SJH Electronic Patient Record (EPR) system, which includes clinical, cardiac, radiology, and laboratory information, as well as discharge summaries. Information was gathered from date of TUDA participation until study completion in June 2013. No exclusion criteria were applied. Information collected included LOS and reason for admission, which was established by extracting relevant information and diagnosis through predefined keywords as outlined above.

Two researchers, Rosaleen Lannon (RL) and Avril Beirne (AB), independently investigated a review of records for reason for ED attendance and hospital admission. Overall, thirty cases were initially reviewed for standardisation of keyword analysis and reason for ED attendance and a further thirty for reason for admission. This review was achieved with a kappa score of greater than 0.8 for inter-rater reliability [24].

Attendance at emergency departments and admissions to other institutions were not captured in this study. Only 1% of the TUDA population studied lived outside the catchment area of SJH; therefore, our data collection captured events for 99% of the study population.

Ethical approval was granted for this study from the St. James’s Hospital/Tallaght University Hospital Joint Research Ethics Committee (REC Reference 5 May 2013 Chairman’s Action).

### 2.4. Statistical Analysis

All parameters were inspected for normality and, if significantly skewed, were appropriately transformed. Normal assumptions for linear regression analysis were observed. Descriptive and comparative analyses were performed for all included participants and for a priori determined subgroups.

Continuous variables are expressed as mean and standard deviations for normally distributed and median and inter-quartiles for non-normally distributed data. Categorical variables are expressed as number of cases and percentages. Nominal or dichotomous variables were compared using chi-squared test, non-normally distributed variables were compared with the Mann–Whitney U-test, and normally distributed continuous variables were compared using Student’s *t*-test. These analyses were run using SPSS v22.0 (SPSS, Inc., Chicago, IL, USA). Statistical significance was accepted when *p* < 0.05.

Survival analysis was performed using R (Bell Technologies, New Jersey, USA) [25]. Cox proportional hazard models were generated for vitamin D as a continuous variable and as a categorical variable based on defined vitamin D concentrations, and were used to evaluate the proposed relationship between vitamin D and ED attendance and hospitalisation. The vitamin D cut-off of >50 nmol/L, defined as sufficient, was based on recommendations from the Institute of Medicine (IOM) [26]. Missing data were dealt with using multiple imputation modelling in multivariate imputation by chained equations (MICE) using R [25].

Interaction terms for cohort and ED attendance and admission status were examined but were not found to be significant in regression analysis.

### 2.5. Covariates

The basic model (Model 1) considered covariates known to effect serum 25(OH)D concentrations, ED attendance, and hospital admission including age, sex, and total number of years in education, Body Mass Index (BMI), vitamin D supplementation and global solar radiation (GSR). The GSR in the month and preceding two months of subject recruitment was used as a surrogate marker of Ultraviolet B (UVB) exposure, which is known to effect serum vitamin D concentrations. GSR represents the total amount of solar radiation (direct beam plus the diffuse component on a horizontal surface) received per unit area per month (MJm^−2^), (http://www.met.ie/about/valentiaobservatory/solarradiation.asp). GSR data were obtained by request from the Irish Meteorological Service.

Model 2 included all variables in Model 1 and variables that were confounders for both physical and cognitive frailty, which are known to affect ED attendance and admission rates: timed up and go (TUG) [27], which is a frequently used screening tool to measure patients gait speed and is associated with a potential risk of falling; Mini Mental State Examination (MMSE) [28], which is a screening tool for cognition and can be used as a marker of cognitive frailty; and finally Instrumental Activity of Daily Living (IADL) [29], a score reflecting a patient’s ability to function independently in their activities of daily living.

Model 3 included all variables from Models 1 and 2 and pre-specified diseases/conditions that might account for ED attendance and admission (i.e., fall in the past year, chronic kidney disease, and history of cancer).

## 3. Results

### 3.1. Baseline Characteristics

Over the mean follow-up of 3.6 years, 1577 (50.9%) TUDA participants attended the ED (Table 1). The median time to first ED attendance was 10.0 months (range 0–51 months) and the median number of attendances was 2.0 (range 1–39 visits). Those participants who attended the ED were older (79.2 vs. 73.7 years), less educated (10.0 vs. 12.0 years), and more likely to be living alone. They also had lower vitamin D concentrations (59.1 vs. 70.6 nmol/L). Those who attended ED were less likely to be taking vitamin D supplementation (62.8%) compared with those who did not attend the ED (67.3%).

The 1269 participants (41%) admitted to hospital were older (80.0 vs. 74.1 years) and more likely to be less educated. Those participants admitted had lower mean vitamin D at baseline, 58.4 nmol/L vs. 69.3 nmol/L. Of those participants admitted, 584 (46%) had 25(OH)D concentration <50 nmol/L and were deemed to have insufficient levels. Again, those who were admitted were less likely to be taking vitamin D supplementation (62.4% compared with those who were not admitted to hospital (66.9%)).

The reasons for first ED presentation and hospital admission are presented in Figure 1 which shows that the most frequent reason for first ED attendance and for hospital admission was due to cardiopulmonary conditions followed by falls and fractures.

### 3.2. Survival Analysis: ED Attendance

Fully adjusted Cox proportional hazard models with vitamin D as the continuous variable showed an inverse association between vitamin D and ED attendances (HR 0.996, 95% CI 0.995–0.998, *p* < 0.001; Table 2). For every 1 nmol/L increase in serum 25(OH)D concentrations, there was a 0.4% reduction in the risk of ED attendance.

Further modelling with vitamin D concentrations presented as a dependent categorical variable continued to show the inverse association between vitamin D and ED attendance. Vitamin D sufficiency (defined as ≥50 nmol/L) was associated with a reduced risk of ED attendance over the follow up period (Relative Risk (RR) 0.76, *p* < 0.001). Models were adjusted for variables known to be associated with vitamin D deficiency and frailty markers.

### 3.3. Survival Analysis: Hospital Admission

In multivariate survival analysis, with Cox proportional hazard modelling, vitamin D, considered as a continuous variable (Table 2), was found to be inversely associated with hospital admission (HR 0.997, 95% CI 0.995–0.998, *p* < 0.001). This association remained in fully adjusted models (HR 0.996, 95% CI 0.994–0.998, *p* < 0.001). Thus, for every 1 nmol/L increase in 25(OH)D, the relative risk of hospital admission was reduced by 0.4%.

Further analysis considering vitamin D as a categorical variable (Table 3) revealed an association between hospital admission and participants with insufficient (<50 nmol/L) vitamin D. Once again, this association was present in fully adjusted models with covariates for both physical and cognitive frailty considered. Thus, for an individual with 25(OH)D concentrations above 50 nmol/L, the relative risk of hospital admission was lower, i.e., 0.77 times that of those with concentrations <50 nmol/L (Figure 2 and Table 3).

Vitamin D concentration was inversely associated with overall (accumulated overall hospital admissions) hospital LOS throughout the study period (β = −0.95, *p* = 0.006), (Table 4 and Figure 3).

## 4. Discussion

This study demonstrated an inverse relationship between vitamin D deficiency and ED attendance, hospital admission rate, and LOS in an older population of community-dwelling adults.

Participants with concentrations of 25(OH)D < 50nmol/L were more likely to attend the ED and be admitted. This relationship remained robust in fully adjusted models accounting for multiple cofounders for vitamin D status, measures of physical and cognitive frailty, and for a number of chronic conditions. Those participants with lower vitamin D who were admitted to hospital were also more likely to have longer hospital LOS compared with vitamin-D-replete participants.

To the best of our knowledge, this is the first study to examine the prospective association between vitamin D deficiency and hospitalisation in a large population of older adults. Earlier studies suggested an association between vitamin D and certain conditions associated with increased resource use, such as chronic conditions, falls [1], frailty [30,31], and fractures [32,33]. Prior studies have suggested that vitamin D deficiency is associated with increased risk of hospital re-admission, nursing home admission, and increased hospital LOS [34,35,36,37].

Whereas some studies have shown an association between vitamin D and conditions leading to admission such as multiple sclerosis and Chronic Obstructive Pulmonary Disease (COPD) [7,38], and infections [4,6,39], this is the only study to date evaluating the relationship between vitamin D and hospital admission rather than specific diseases or illnesses.

A number of reports indicate a high prevalence of vitamin D deficiency in European countries, leading to concerns about the associated health risks [40]. An Irish study reported the prevalence of vitamin D deficiency (<30nmol/L) as 13% and deficiency status was more prevalent in those aged over 80 years [41]. Older compared to younger adults are at higher risk of vitamin D deficiency due to reduced sun exposure, lower capacity to synthesise vitamin D in skin, and lower dietary intakes of dietary vitamin D. Our previous research from the TUDA study showed that vitamin D supplement use was the most important determinant of vitamin D status in the TUDA cohort, whilst vitamin-D-fortified food and spending time in the sun, even in the oldest, were effective in improving 25(OH)D concentrations [23]. We also previously showed that vitamin D deficiency was associated with a more pronounced pro-inflammatory status, including higher Interleukin (IL)-6 and C-reactive protein (CRP) in older Irish adults [42].

Vitamin D deficiency occurs commonly in older adults and, given its adverse effects on immune functioning, has received increased attention during the time of the COVID-19 pandemic. Although to date there are no published intervention trials in relation to SARS-CoV-2, it is possible that better vitamin D status may be beneficial in terms of the host response to initial infection or in determining health outcomes should an older person become infected with corona virus [43,44].

Although a large population of older adults was included in this analysis, with a 3.6 year follow-up, and multiple potential confounders were considered, there are some potential limitations to our study. Participants were originally recruited to cohorts with underlying diseases of cognition and bone health and, therefore, the results may not be fully representative of a general older population. Data were only available for our own institution so it is possible that participants could have attended another facility and these data were not available for this study. As only approximately 1% of participants lived outside our catchment area, this possibility is unlikely.

Finally, frailty is an important factor potentially contributing to the reason for ED attendance. While we did not use a specific frailty score or tool in this study, as there is no consensus on the best frailty index at present, frailty was considered and measured using markers of physical and cognitive frailty as previously outlined and which were included as confounders in this study’s analysis.

## 5. Conclusions

Our study’s findings suggest vitamin D deficiency is associated with increased hospitalisation by older adults independent of markers of physical and cognitive frailty. This prospective association strengthens the need to consider vitamin D as an independent modifiable factor in ED attendance and hospital admission rates and the potential need for supplementation in older adults deficient in vitamin D to bring concentrations >50 nmol/L. The need for further evaluation of current recommendations in relation to vitamin D supplementation, with consideration of the effects of deficiency beyond bone health, is warranted and should focus on randomised controlled trials.

## Figures and Tables

**Figure 1 nutrients-13-00616-f001:**
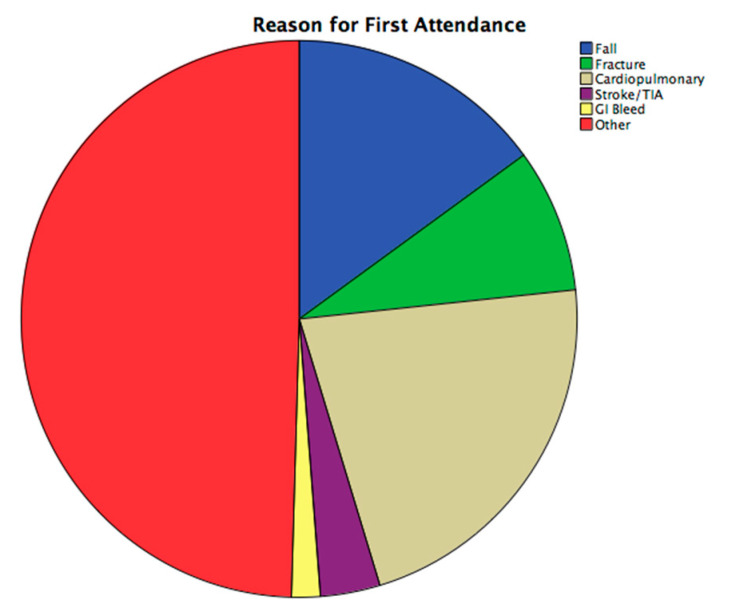
Reason for first emergency department (ED) attendance and first hospital admission.

**Figure 2 nutrients-13-00616-f002:**
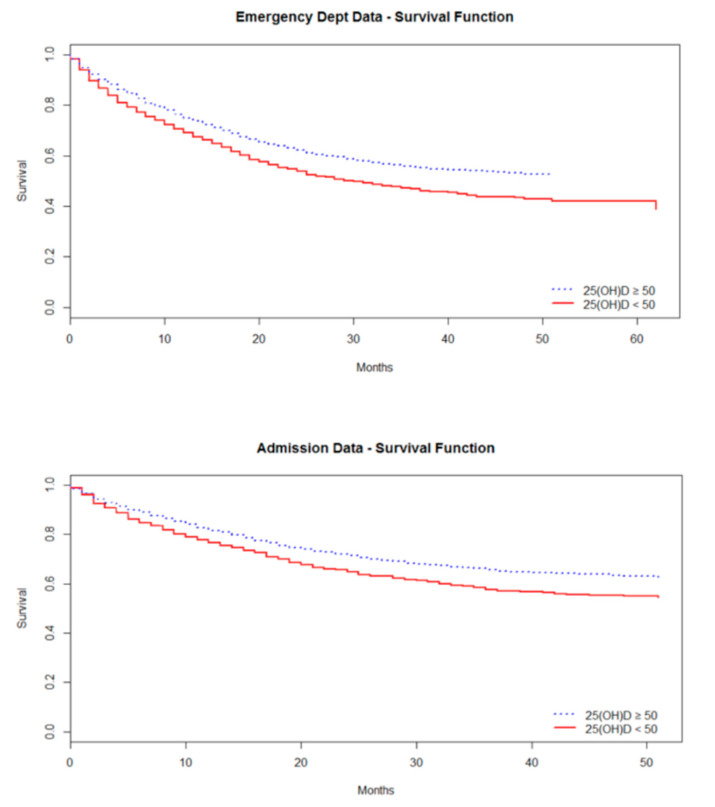
Survival curve 25 hydroxyvitamin D >/< 50 nmol/L: ED attendance and hospital admissions.

**Figure 3 nutrients-13-00616-f003:**
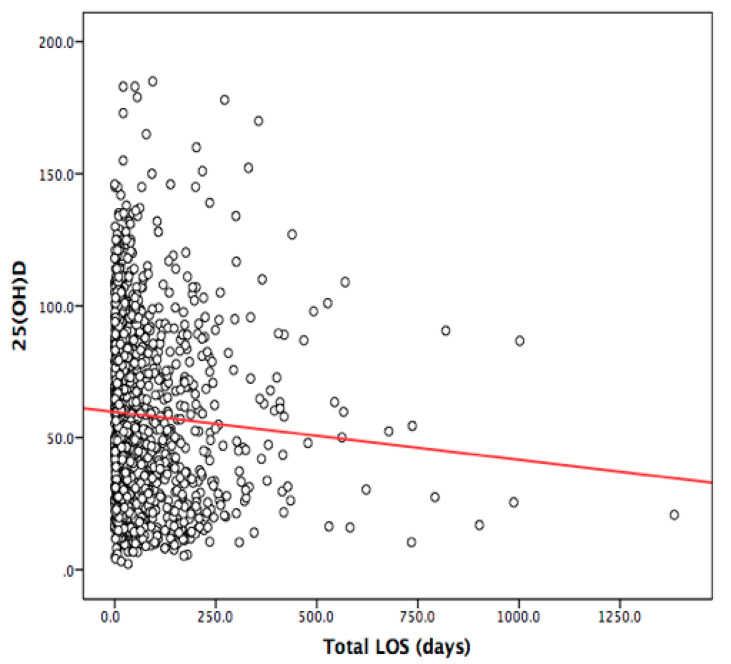
Scatter plot of the association between total hospital length of stay (LOS) and 25(OH)D concentration.

**Table 1 nutrients-13-00616-t001:** Baseline characteristics by emergency department (ED) and admission status.

Baseline Characteristic	ED Attenders (*n* = 1557)	ED Non-Attenders (*n* = 1536)	*p*-Value	Admitted (*n* = 1269)	Not Admitted (*n* = 1824)	*p*-Value
Cognitive Cohort (%)	72.7	37.0	<0.001 *^,b^	76.6	39.9	<0.001 *^,b^
Bone Cohort (%)	27.3	63.0	<0.001 *^,b^	23.4	60.1	<0.001 *^,b^
Age (years), Mean (SD)	79.2 (±7.8)	73.7 (±8.2)	<0.001 *^,a^	80.0 (±7.6)	74.1 (±8.3)	<0.001 *^,a^
Sex, female (%)	70.0	79.9	<0.001 *^,b^	68.5	79.8	<0.001 *^,b^
Education (years), Mean (SD)	10.0 (±2.5)	12.0 (±3.5)	<0.001 *^,c^	10.0 (±2.5)	11.0 (±3.4)	<0.001 *^,c^
Living Alone (%)	43.7	36.1	<0.001 *^,b^	45.0	36.4	<0.001 *^,b^
Current Smoker (%)	13.0	12.6	0.747 ^b^	12.9	12.8	0.912 ^b^
Previous Smoker (%)	43.9	37.7	<0.001 *^,b^	44.5	38.2	<0.001 *^,b^
Current Alcohol (%)	52.7	61.4	<0.001 *^,b^	51.0	61.2	<0.001 *^,b^
Previous Alcohol (%)	22.9	14.6	<0.001 *^,b^	24.2	15.0	<0.001 *^,b^
25(OH)D (nmol/L), Mean (SD)	59.1 (±33.3)	70.6 (±31.9)	<0.001 *^,c^	58.4 (±33.6)	69.3 (±32.0)	<0.001 *^,c^
25(OH)D (nmol/L), Median (Range)	55.0 (2.2–185.0)	71.9 (3.7–288.0)	<0.001 *^,c^	53.9 (2.2–185.0)	70.4 (3.7–288.0)	<0.001 *^,c^
Vitamin D supplementation (%)	62.8	67.3	0.012 *^,b^	62.4	66.9	0.012 *^,b^
MMSE, Median (Range)	27.0 (6.0–30.0)	27.0 (5.0–30.0)	0.074 ^c^	27.0 (6.0–30.0)	27.0 (5.0–30.0)	0.033 ^c^
HADS, Median (Range)	2.0 (0–21.0)	2.0 (0–21.0)	0.574 ^c^	2.0 (0–21.0)	2.0 (0–21.0)	0.309 ^c^
CES-D, Mean (SD)	6.6 (±7.4)	5.9 (±7.1)	0.014 *^,a^	6.5 (±7.3)	6.0 (±7.2)	0.056 ^a^
TUG (seconds), Median (Range)	16.0 (4.0–140.0)	12.0 (3.0–68.0)	<0.001 *^,c^	16.0 (4.0–140.0)	12.0 (3.0–68.0)	<0.001 *^,c^
BMI (kg/m2), Mean (SD)	26.9 (±5.3)	26.6 (±5.3)	0.109 ^a^	26.9 (±5.4)	26.6 (±5.3)	0.875 ^a^
Waist–Hip Ratio, Mean (SD)	0.9 (±0.1)	0.9 (±0.1)	0.022 ^a^	0.9 (±0.1)	0.9 (±0.1)	0.68 ^a^
IADL, Mean (SD)	22.3 (±4.6)	23.1 (±4.6)	<0.001 *^,a^	22.4 (±4.7)	23.0 (±4.6)	<0.001 *^,a^
PSMS, Median (Range)	24.0 (10.0–24.0)	24.0 (9.0–24.0)	0.292 ^c^	24.0 (10.0–24.0)	24.0 (9.0–24.0)	0.203 ^c^
Hypertension (%) **	83.1	75.5	<0.001 *^,b^	83.2	76.6	<0.001 *^,b^
Vascular Disease (%) ^^^	30.4	26.6	0.019 *^,b^	29.6	27.8	0.267 ^b^
Diabetes (%)	11.5	8.4	0.003 *^,b^	11.4	9.0	0.026 *^,b^
Atrial Fibrillation (%)	16.0	14.6	0.270 ^b^	15.7	15.0	0.602 ^b^

^a^ Independent t-test; ^b^ Chi-Square Test; ^c^ Mann–Whiney U-Test; * Statistically significant result; ^^^ Vascular disease: self-reported history of stroke, transient ischaemic attack, peripheral vascular disease and/or myocardial infraction; ** Hypertension: self-reported history, medication or Blood Pressure (BP) >140/90 mmHg; BMI: Body Mass Index, CES-D: Centre for Epidemiologic Studies Depression Scale, HADS: Hospital Anxiety and Depression Scale, IADL: Instrumental Activities of Daily Living Scale, MMSE: Mini Mental State Examination, PSMS: Physical Self-Maintenance Scale, SD: Standard Deviation, TUG: Timed up and Go.

**Table 2 nutrients-13-00616-t002:** Cox proportional hazard 25 hydroxyvitamin D and emergency department (ED) attendance (vitamin D as a continuous time dependent variable).

**ED Attendance**	**β Coefficient**	**HR**	**95% CI**	***p*** **Value**
Model 1	−3.570	0.996	0.995–0.998	<0.001 *
Model 2	−3.644	0.996	0.994–0.998	<0.001 *
Model 3	−3.868	0.996	0.994–0.998	<0.001 *
**Hospital Admission**	**β Coefficient**	**HR**	**95% CI**	***p*** **Value**
Model 1	−3.485	0.997	0.995–0.998	<0.001 *
Model 2	−3.768	0.996	0.994–0.998	<0.001 *
Model 3	−4.035	0.996	0.994–0.998	<0.001 *

Model 1: Age, sex, living alone, total education, vitamin D supplementation, GSR, and BMI; Model 2: Model 1 and TUG, MMSE, and IADL; Model 3 ED: Model 2 and fall in the last year and history of cancer; Model 3 admission: Model 2 and history of cancer, CKD, and fall in the past year; * Statistically significant value; BMI: Body Mass Index, CKD: Chronic Kidney Disease, GSR: Global Solar Radiation, HR: Hazard Ratio, IADL: Instrumental Activities of Daily Living Scale, MMSE: Mini Mental State Examination, TUG: Timed Up and Go, CI: Confidence Interval.

**Table 3 nutrients-13-00616-t003:** Cox proportional hazards 25 hydroxyvitamin D >/<50nmol/L: ED attendance and hospital admission (vitamin D as a categorical variable).

**ED Attendance**	**β Coefficient**		**RR**	***p*** **Value**
Model 1	−0.2519		0.77	<0.001 *
Model 2	−0.2667		0.77	<0.001 *
Model 3	−0.2740		0.76	<0.001 *
**Hospital Admission**		**β Coefficient**	**RR**	***p*** **Value**
Model 1	−0.2521		0.77	<0.0001 *
Model 2	−0.2574		0.77	0.0004 *
Model 3	−0.2638		0.77	0.0003 *

Model 1: Age, sex, living alone, total education, vitamin D supplementation, GSR, and BMI; Model 2: Model 1 and TUG, MMSE, and IADL; Model 3: Model 2 and fall in the last year and history of cancer; * Statistically significant value; BMI: Body Mass Index, GSR: Global Solar Radiation, IADL: Instrumental Activities of Daily Living Scale, MMSE: Mini Mental State Examination, TUG: Timed Up and Go, RR: Relative Risk.

**Table 4 nutrients-13-00616-t004:** Association between total hospital length of stay (LOS) (log transformed) and 25(OH)D concentration (log transformed).

Length of Stay	β Coefficient	95% CI	*p* Value
Model 1	−0.74	−0.27–−0.03	0.015 *
Model 2	−0.95	−0.33–−0.56	0.006 *

Model 1: Age, sex, living alone (Y/N), education, GSR, vitamin D supplementation, BMI, and time to assessment; Model 2: Model 1 and TUG, MMSE, and IADL; * Statistically significant value; IADL: Instrumental Activities of Daily Living Scale, MMSE: Mini Mental State Examination, TUG: Timed Up and Go

## Data Availability

Publicly available datasets were analysed in this study. This data can be found here: https://www.ucd.ie/jingo/database/tuda/.

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
