# Peer review of "Vitamin D and Hospital Admission in Older Adults: A Prospective Association"

_nutrients, 2021, doi:10.3390/nu13020616_

Round 1
Reviewer 1 Report
According to Table 1, >60% of the patients had taken Vit D - supplements:
Please indicate in addition the Vit D dose range.
Line 258 pp: Please discuss the dosing of Vit D-supplements: enough - or in (de)crease dosings?
How long did patients take the supplements - did they really take them??
Should dosing generally be in(de)creased: yes or no? Precautions?
Author Response
Many thanks for your comments and suggestions.
The supplementation of Vitamin D was not instituted as part of this study. Participants self-reported their vitamin D supplements with doses ranging from 100IU to 800IU per day.
Compliance was not recorded as we did not initiate supplementation therefore I can not accurately comment on this or the duration of therapy prior to recruitment, unfortunately.
So this leaves it difficult to say if their doses should be adjusted. I can recommend that those deficient need depletion and avoidance of supra- supplementation?
Many thanks,
Avril
Reviewer 2 Report
The manuscript determines the relationship of vitamin D status in older adults with hospital admission and emergency department attendance.
Authors found inverse associations of vitamin D with hospital admission. According to the author, vitamin D supplementation will improve the disease states of cognition, bone health, and hypertension.
The author presents a nice piece of work that will recommend vitamin D supplementation in older adults. In my opinion, this work is acceptable.
- The title and abstract reflect the content of the work.
- The introduction section requires some improvement. Please add the role of vitamin D in cognition, bone health, and hypertension.
- The authors have performed good statistical analysis to find an association between vitamin D and cognition, bone health, and hypertension in older adults.
- Discussion and explanations of the obtained results were made correctly.
- The selected topic is interesting. The work is properly documented.
Author Response
Many thanks for your review. I have made the suggested improvement to the introduction in the manuscript.
Thanks, Avril Beirne